# Evaluating the Effects of a Progressive Kinesiotaping Treatment Protocol on Chronic Low Back Pain in Women Using Electroencephalography

**DOI:** 10.3390/jfmk10030338

**Published:** 2025-09-03

**Authors:** Ana Carolina F. T. Del Antonio, Tiago T. Del Antonio, Marieli Ramos Stocco, Alex Silva Ribeiro, Nelson Morini Junior, Adriana Bovi, Claudia S. Oliveira, Deise A. A. P. Oliveira, Dante B. Santos, Iransé Oliveira-Silva, Rodrigo F. Oliveira, Luís V. F. Oliveira, Luciana Prado Maia, Rodrigo A. C. Andraus

**Affiliations:** 1Physhiotherapy Department, The State University of Northern Paraná (UENP), Jacarezinho 86400-000, PR, Brazil; acftsunoda@uenp.edu.br (A.C.F.T.D.A.); tdelantonio@uenp.edu.br (T.T.D.A.); marieli.stocco@uenp.edu.br (M.R.S.); 2Faculty of Sports Sciences and Physical Education (FCDEF), University of Coimbra, CIPER, 3040-248 Coimbra, Portugal; alex-silvaribeiro@hotmail.com; 3Master Degree and PhD Graduate Program in Human Movement and Rehabilitation, Evangelical University of Goiás (UniEVANGELICA), Anápolis 75083-515, GO, Brazil; nelson.morini@ulife.com.br (N.M.J.); adrianabovil@gmail.com (A.B.); csantos.neuro@gmail.com (C.S.O.); deisepyres@gmail.com (D.A.A.P.O.); dantebsantos@gmail.com (D.B.S.); iranse.silva@unievangelica.edu.br (I.O.-S.); rodrigofranco65@gmail.com (R.F.O.); oliveira.lvf@gmail.com (L.V.F.O.); 4Dentistry Graduate Program, Evangelical University of Goiás (UniEVANGÉLICA), Anápolis 75083-515, GO, Brazil; lucianapmaia@gmail.com

**Keywords:** kinesiotaping, electroencephalography, low back pain

## Abstract

**Objectives**: The central nervous system plays a fundamental role in chronic pain; however, its behavior in this condition remains unclear, especially when associated with interventions such as kinesiotaping (KT). This study aimed to analyze the effects of KT on the somatosensory cortex of women with chronic low back pain. **Methods**: This case series involved 15 women with chronic low back pain. Participants underwent a progressive-tension KT protocol for 8 weeks, and electroencephalogram recordings were performed in two positions, namely sitting and standing while load bearing (10% of body weight), in the first and eighth weeks. The following instruments were employed: Oswestry lumbar disability index, fear avoidance beliefs questionnaire, and the numerical pain intensity scale. **Results**: All participants showed significant pain improvement and a reduction in Oswestry disability index scores from moderate to minimal. Additionally, activity in the alpha band within the somatosensory cortex and insula (central region—represented by the electrode Cz) decreased. This was confirmed by reduced power spectral density, indicating diminished cortical activity in these regions. **Conclusions**: KT positively affects women with chronic low back pain, providing pain reduction and improved functional capacity, as indicated by the fear avoidance beliefs questionnaire and numerical pain intensity scale. Moreover, KT reduces cortical activity in the somatosensory cortex, which is related to the progression of painful sensations, described above after the intervention.

## 1. Introduction

The International Association for the Study of Pain (IASP) defines pain as an unpleasant sensory and emotional experience associated with, or similar to, actual or potential tissue damage [1]. It is a distressing experience with an essential protective function for survival. Moreover, pain is a personal and subjective experience influenced by genetic, sensory, psychological, emotional, cultural, and social factors [2].

Factors associated with the pain process are widely studied to better understand their actual effects on the body. However, because pain is a multidimensional concept, chronic pain generates confusion in this process [3,4,5]. Theoretically, chronic pain is defined as pain in the previous 3 months and/or episodes of pain for at least half of the days in the past 6 months [6,7].

Information regarding the state of the tissues coming from peripheral receptors (without acute tissue damage) and the central processing of this information (perceived pain reflecting a threat to the tissues) is part of the interpretation of the characteristics of this clinical condition [2]. Therefore, pain can go from an appropriate protective response to becoming overprotective, or it can result from a heightened state of sensitivity unrelated to tissue pathology. The persistence of the heightened sensitivity state in central nervous system (CNS) pain pathways may characterize chronic low back pain [2,3,4,5,6,7,8].

With advances in techniques to better understand chronic pain mechanisms, studies have analyzed cortical activity and identified that patients with chronic pain present alterations in brain function and structure compared to pain-free individuals. These alterations are related to pain persistence, demonstrating a positive association between pain duration and brain activity changes [9,10].

Importantly, pain is highly prevalent in the lumbar spine. Clinically, it has a varied and complex etiology, prompting a range of studies aimed, among other goals, at improving the quality of life of affected individuals given its high prevalence, affecting approximately 60% to 80% of the economically active population [3]. A study which evaluated 195 countries in relation to incidence, prevalence, and years lived with functional disability demonstrated that low back pain is the main cause of productivity loss worldwide when measured in years, and the primary cause of years lived with disability in 126 countries [4].

With the advancement of techniques to better understand the mechanisms of chronic pain, studies have analyzed cortical activity and identified that patients with chronic pain have changes in brain function and structure compared to patients without pain. These changes are related to the persistence of pain, demonstrating a positive association between the duration of pain and changes in brain activity [9,10].

Kinesiotaping (KT) is a technique that has become widely used globally due to its diverse application possibilities. Currently, several studies on KT have been conducted to verify its actual physiological effects [11,12]. Although its mechanisms are not yet fully understood, local sensory-motor feedback and pain reduction may be explained by elevation of the skin and subcutaneous space. This results in reduced activation of pain receptors and possible activation of the descending inhibitory system because individuals with chronic pain have functional abnormalities in brain areas associated with attention, salience, and pain processing and modulation [11]. This mechanism can also be justified by the gate control theory of pain, whereby tactile stimulation of the KT would reduce the afferent signal from large-diameter non-nociceptive fibers, resulting in relief of the painful symptom [8,13,14].

The CNS is crucial in chronic pain [15,16]. Therefore, brain imaging techniques such as functional magnetic resonance imaging, cranial tomography, and electroencephalography (EEG) may help in understanding the functional reorganization of the brain network in relation to chronic pain [17]. Specifically, EEG is increasingly being used to determine parameters that can help track brain modulation during peripheral interventions. Nonetheless, most EEG studies are conducted mainly in the context of experimental pain. For instance, studies on chronic pain suggest that alpha frequency power decreases in relation to the pain stimulus [18,19,20,21,22], and beta frequency power is more closely related to pain intensity [22,23]. Notably, a study demonstrated that adults with chronic pain exhibited altered EEG patterns compared with healthy controls, including a shift towards increased power in higher frequency bands [24,25,26].

These findings encourage the use of EEG in low back pain research. Identifying the potential changes occurring in the central nervous system during non-drug interventions, such as in the present study, is warranted. Therefore, this study aimed to analyze the effects of KT on somatosensory cortex function, as measured by EEG, and on pain intensity, disability, and fear avoidance beliefs in women with chronic low back pain.

## 2. Materials and Methods

This was a case series study involving 15 participants (all women). The sample size was calculated using GPower software version 3.1.9.4, considering a power of 80%, alpha of 0.05, and effect size of 0.78 [27]. Participants were aged between 28 and 50 years.

This study was approved by the Research Ethics Committee of Universidade Pitágoras Unopar Anhanguera under protocol number 3.059.113.

### 2.1. Participants

This study included 15 women with chronic low back pain, confirmed by the Oswestry lumbar disability index (moderate/severe intensity), with pain persisting for at least 3 months or recurring in the last 6 months [1,2,3,4].

The following were considered eligibility criteria for this study:

(a) Not being in menopause,

(b) No history of back surgery,

(c) No confirmed diagnosis of any orthopedic changes in the lumbar spine (e.g., herniated disks),

(d) Pain intensity score greater than 5 points pre-intervention.

Women were considered ineligible if they:

(a) Missed one or more sessions during the KT protocol,

(b) Ingested psychotropic or narcotic medications during the intervention period,

(c) Underwent physiotherapeutic treatment during the protocol,

(d) had severe central and peripheral neurological, psychiatric, rheumatological, and cardiac diseases. 

Individuals who did not meet the eligibility criteria were referred to specific therapeutic groups available at the higher education institution where this study was carried out.

All participants were informed about the research procedures and were asked to sign the informed consent form.

### 2.2. Instruments

Initially, the researcher completed an identification form with personal and anthropometric data for later characterization of the sample and asked the participants to fill out the following questionnaires: Oswestry lumbar disability index (ODI), fear avoidance beliefs questionnaire (FABQ) [28], and the numerical pain intensity scale (NPRS).

The participants were given guidelines regarding the intervention period, such as not using analgesic pain medication during the protocol, maintaining KT for a maximum of 5 days, and not drinking alcohol or caffeine for 12 h before the EEG was taken.

The NPRS [29] consists of a numerical sequence from 0 to 10, ranging from “no pain” to “worst possible pain.” Pain intensity was measured every week of the protocol before KT was applied.

The ODI, validated for Portuguese [30,31] in 2007, is designed to assess disability related to chronic pain. This instrument consists of 10 sections, each with six alternative answers. Moderate/severe disability was used as the eligibility criterion for this study.

The FABQ is dedicated to the cognitive–behavioral analysis of fear, beliefs, and avoidance behaviors in individuals with chronic low back pain in relation to the domains of physical activity and work. It consists of 16 self-reported items and is subdivided into FABQ-Phys, which addresses beliefs related to occupational activities, and FABQ-Work, which addresses beliefs related to work [28].

### 2.3. Procedures

The protocol lasted for 8 weeks. In the first and last weeks, the participants received KT, filled in the questionnaires, and underwent an EEG. From the second to the seventh weeks, they received the application of the bandage, and in the eighth week, the procedures were repeated as described below.

Using a tape measure, the researcher measured the distance between the transverse process of the twelfth thoracic vertebra and centimeters below the posteroinferior iliac spine (EIPI), so that during the protocol these anatomical points could serve as a reference for the initial size of the beige Therapy Tex^®^ taping.

KT was applied to the lumbar spine in an I-shape (bilaterally), with the upper anchor placed on the transverse process of T12 with the patient seated and the trunk in a neutral position, followed by slight trunk flexion and placement of the lower anchor 5 cm below the EIPI, immediately after cleaning the application site with 70% alcohol (Figure 1).

During the subsequent 7 weeks, individuals received KT at home (once a week), following the progressive tension protocol by reducing the size of the tape while maintaining the anatomical points as anchoring references, based on the following formula:A (distance between anatomical points) × 20% (half of the product’s total tension capacity) = X (product)X/B (number of weeks of treatment) = Y (weekly reduction in bandage size)

Re-evaluation took place in the eighth week, during which the EEG was performed in the same posture and by the same technician who carried out the initial recording, along with the completion of the questionnaires and NPRS as previously described.

In the first and last sessions of the protocol, participants underwent an EEG with a BWII EEG machine (Neurovirtual, Doral, FL, USA), which was provided by Public Intermunicipal Health Consortium of the Northern Pioneer Region. The International 10–20 System was used to collect the data and position the EEG leads.

The participant took the test in two positions:

Position 1—Seated, with arms relaxed over the legs, knees bent at 90°, feet flat on the floor, and torso leaning back against the chair. Participants were required to avoid long blinks during the 5 min they would remain in this position (30 s of eyes open, 30 s to 2 min of eyes closed, and 2 to 5 min of eyes open fixed on a point marked on the wall), and the commands for opening and closing the eyes were made by a beep previously recorded by the researcher and explained to participants.

Position 2—Standing while load bearing; the participant picked up a basket representing 10% of their body weight [31], with elbows bent and eyes open and fixed on the previously marked point at eye level, for a maximum time of 5 min or 300 s in this position the limit was indicated by the participant when the pain level became moderately bothersome, according to the numerical descriptive tool. At the beginning and end of each posture, the researcher marked on the evaluation form the places where the participant felt the most discomfort during data collection.

Here, the EEG data collected during the first 30 s with eyes open (baseline) and the last 3 min or 180 s with eyes open were considered for later analysis in the resting posture. As the patients in posture 2 were asked to maintain the load for a maximum of 5 min, the shortest time, approximately 1 min (60 s), was obtained by patient 14. Thus, the EEG data from the first 60 s of recording were retained for analysis while standing with dynamic load bearing, as shown in Figure 2.

### 2.4. EEG Data Recording and Analysis

Pre-processing of the EEG signals was performed in three steps according to Segning et al. [32,33]—removal of EEG artifacts, selection of EEG frequency bands, and normalization of artifact-free EEG signals in two sub-steps (min–max and baseline normalization). The signals for five electrodes, including T3, C3, Cz, C4, and T4, were of interest (Figure 3) and all pre-processing steps were carried out using MATLAB (version r2022a) analysis [The Math Works Inc., Natick, MA, USA].

The EEG data analysis for both postures (standing and sitting) was performed only in the first 60 s of the posture to facilitate data collection and analysis.

The first stage of pre-processing involved removing artifacts due to blinking or eye movement, EEG, and electromyography signals. An outlier detection and replacement filter was used [32,33]. For this purpose, outliers were defined as EEG values more than 1.5 interquartile ranges above the upper quartile (75%) or below the lower quartile (25%) and detected using the quartile location method. The linear interpolation method of non-discrepant neighboring values was then used to replace the detected discrepant values.

The second stage of preprocessing involved EEG frequency sub-band selection for objective analysis, including alpha (8–12 Hz) and beta (13–30 Hz). Notably, previous studies have reported the involvement of motor cortex areas in the nociception process [34]. Thus, electrodes positioned bilaterally in the cortical motor regions (T3/T4, C3/C4) and in the central region of the cortex (Cz) were analyzed. Finally, a fifth order Butterworth IIR bandpass filter with zero phase filtering was applied to all selected frequency bands [35].

### 2.5. Normalization of EEG Signals

The third step involved the adjustment of EEG data intervariability within each frequency band by scaling EEG data using min–max normalization. This step involves placing the EEG data in the range between 0 and 137. Min–max normalization performs a linear transformation on the original data values, preserving the relationships between them [33,34].

The next sub-step was baseline normalization, which rescales the min–max normalization values by the weight of each. This was implemented by dividing each min–max normalized EEG value by a selected reference. The reference was defined as the average of the min–max EEG values in the baseline condition for the EEG frequency band considered [33,34].

For EEG signals, in the resting posture, the reference was taken as the first second of EEG collection for each patient, while for dynamic posture normalization, the reference was the initial 30 s of EEG recordings with eyes open. The subsequent step involved the power spectral density (PSD) calculation using 1 s with 50% overlapping sliding windows [34,35].

### 2.6. Calculation of PSD

PSD is calculated as a correlate of the degree of activity of the target neuronal population [36]. A high PSD indicates an increase in neuronal activity. Conversely, a decrease in neuronal activity is reflected in a diminished PSD. There are many techniques for assessing PSD. Here, the PSD of each EEG frequency band was estimated using the non-parametric method based on fast Fourier transform (Figure 3). Notably, PSD is a measure of the energy contained in the frequency bands of a signal and is one of the classic tools for analyzing biomedical signals [37,38,39].

PSD was calculated as the squared magnitude of the Fourier transform of the EEG frequency band of interest asPSD(f) = |X(f)|2(1)
where f represents the EEG frequency band considered, i.e., alpha or beta, and X(f) is defined by the following equation [39]:X(f) = ∑x(n)e^(−j2πfT_s_)

Here, n, x(n), and T_s_ represent the total number of points (the length of the digital data within each sliding window), EEG signal considered, and sampling interval, respectively.

The final stage of signal processing involved calculating the relative PSD in percentage (%), representing the normalization of the PSD in each frequency band by the total PSD [40].

### 2.7. Statistical Analysis

The normality of the data was confirmed by the Shapiro–Wilk test. Mean and standard deviation values were used for descriptive analysis. A paired *t*-test was used to compare the relative alpha and beta PSD of the two interventions (pre and post) using the T3, C3, Cz, C4, and T4 electrodes, as well as the pre- and post-intervention values of the NPRS, ODI, and FABQ-Brazil.

The effect size of the variables studied was verified, with values equal to or greater than 0.80, between 0.40 and 0.79, and equal to or less than 0.39 representing large, moderate, and small effect sizes, respectively, according to Hedges’ g. The significance level was set at *p* < 0.05, and all statistical analyses were carried out using SPSS version 24.

## 3. Results

The study sample consisted of 15 women, with a mean age of 36.33 ± 7.47 years and a body mass index of 23.95 ± 2.88 kg/m^2^. Regarding pain intensity, all participants reported scores greater than 7 on the baseline NPRS. Following the intervention, pain scores ranged between 0 and 5. A comparison of the mean pain intensity before and after the intervention revealed a statistically significant difference (*p* = 0.0001) and a large effect size of 0.90, as shown in Table 1.

Table 2 presents the data relating to ODI and FABQ at the pre- and post-intervention stages. Pain intensity decreased from 7.40 ± 0.50 to 1.26 ± 2.05. There was also a reduction in physical disability according to the ODI from 36.80 ± 7.58, moderate disability, to 17.53 ± 7.64, indicating minimal disability. As for the FABQ-Work, significant improvement was observed, with scores reducing from 32.71 ± 18.46 to 23.71 ± 18.59.

The shortest recording time was observed in participant P14; therefore, only the first minute (60 s) of the EEG recording was considered to maintain uniformity in the data analysis.

### PSD Relative

The relative PSD of the alpha and beta frequency bands was analyzed for five electrodes, including T3, C3, Cz, C4, and T4.

In both pre- and post-intervention conditions, only Cz demonstrated a significant difference in the alpha band, showing a reduced relative PSD post-intervention in the resting position (Figure 4).

## 4. Discussion

The primary objective of this study was to determine the effect of KT over an 8-week period on the somatosensory cortex of women with chronic low back pain. This is a contemporary and innovative study, and among the few seeking to understand the potential relationships between KT effects and its influence on the somatosensory cortex.

Regarding pain, studies suggest that the prevalence of chronic low back pain is approximately six times higher in women than in men, which may be associated with hormonal and gestational factors and excessive workload, as well as depressive factors and anxiety disorders. Interventions alleviating painful symptoms can directly affect women’s quality of life [41,42]. This corroborates our findings in women, who exhibited a decrease in the average pain score on the NPRS from 7 to 1 at the beginning and end of the protocol, respectively, demonstrating an improvement in the functional capacity of the target population of this study.

When comparing the use of KT in women with chronic low back pain, pain improvement can be explained by the physiological mechanisms of bandage application. Specifically, the increase in subcutaneous space with skin lifting, caused by the application of the bandage with tension, would result in less activation of pain receptors and, consequently, activate the descending inhibitory system [42,43,44]. This fact corroborates the results of Mohamed et al. [44] who, following KT application, observed a reduction in the pain scores of the volunteers.

Chronic low back pain is strongly associated with functional disability [44]. Consistent with the results of Elabd and Elabd [45,46], who used KT combined with stabilization exercises for the treatment of chronic low back pain, we found an improvement in the patients’ lumbar disability. This information suggests the relevance of physiotherapeutic interventions that can improve aspects such as pain and disability, corroborating our findings, in which functional disability of the lumbar spine, according to the ODI, was reduced from moderate to minimal after the use of a relatively simple intervention. Moreover, it reinforces that, in addition to KT, it is important to combine exercises and techniques that complement the patient’s recovery process.

A significant difference was found in the FABQ-Work domain, corroborating the findings of Silva et al. [46,47,48] who, in a retrospective cross-sectional study, observed that exercise practices or habits beyond physiotherapeutic care can improve work-related fear and avoidance scores. Additionally, they suggested that an active posture for the patient allows them to cope with pain and daily activities, leading to a reduction in fear and recovery, despite some level of pain.

Regarding our EEG results, studies have shown that alpha waves, already recognized in the literature for exhibiting greater oscillation in patients with chronic pain, also present greater oscillation compared to healthy patients with chronic pain [49]. Thus, the reduction in alpha wave activity in our study points to a possible efficacy of KT as a treatment, especially when considering the dimension of pain that extends beyond perception to its processing in the somatosensory cortex. This finding can be justified by the viscoelastic properties of the skin, whereby the mechanical force applied by KT can cause the skin to crease, reducing pressure on the cutaneous mechanoreceptors, which in turn activate large-diameter fibers, including A-beta, C, and A-delta fibers. These fibers, located in joints, muscles, tendons, and skin in painful areas, activate inhibitory interneurons to block pain signals conducted by A-delta and C fibers [48,49].

The electrode that identified this change in our study (Cz) is located in the central region of the head and, together with other electrodes, captures electrical responses in regions involving the interpretation of pain. The two main ascending pain pathways include the anterior cingulate cortex and the insular cortex, strategically subdivided into three regions: the posterior insula, which processes all sensory information (olfactory, taste, auditory, somatosensory, and pain); the ventral area, which is involved in emotional processing; and the dorsal area, which is involved in cognitive processing [41,42]. Thus, the reduced cortical activity in these regions suggests the positive influence of treatment on the interpretation, processing, and modulation of pain. Importantly, this fact has been little investigated in the literature, which makes it difficult to find studies that corroborate this information.

Exploring the correlation of specific regions of the somatosensory cortex associated with pain with a larger number of patients and a methodological design that ensures the methodological quality of the results is warranted.

The limitations of this study include the absence of a control group, which, among other factors, makes it difficult to compare the effects of the protocol and the assessment of potential influences from external factors. Additional limitations are the use of a single KT application method, the small sample size, and methodological weaknesses of low control of confounding factors, such as hormonal fluctuations and levels of daily activity.

## 5. Conclusions

Our study demonstrated a significant reduction in alpha wave activity in the electrode region, corresponding to the central region of the cerebral cortex, as shown by the PSD analysis of the EEG. Although the study presented some limitations, the difficulty in delimiting sensitivity analysis in EEG data may casually infer the quality of the results. Nevertheless, when considering clinical effects such as pain reduction, the comparison of pre- and post-intervention data revealed that the reduction in neuronal activity in the central region cortex was accompanied by a significant decrease in pain and, consequently, an improvement in the functional capacity of the volunteers.

## Figures and Tables

**Figure 1 jfmk-10-00338-f001:**
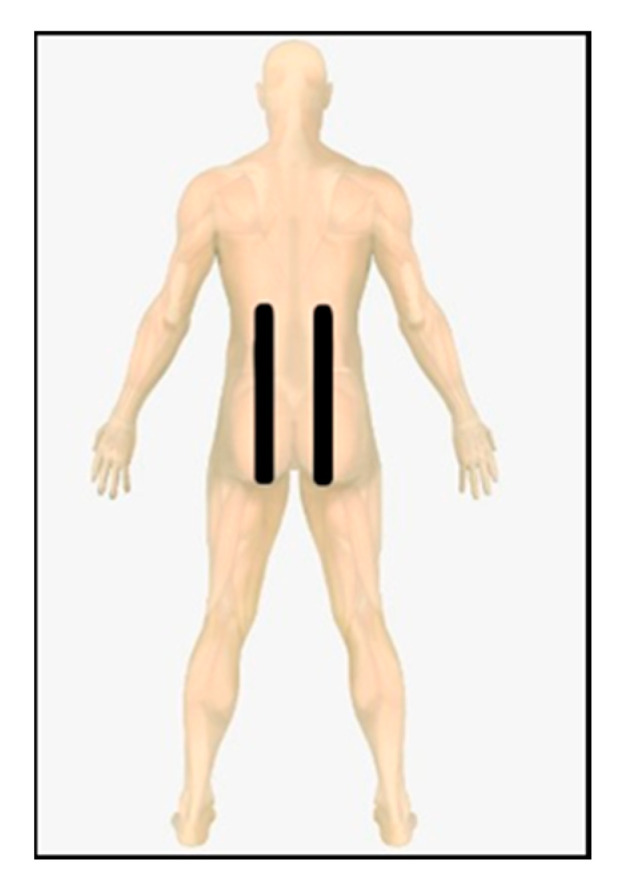
Application of Kinesiotaping. Source: Itself.

**Figure 2 jfmk-10-00338-f002:**
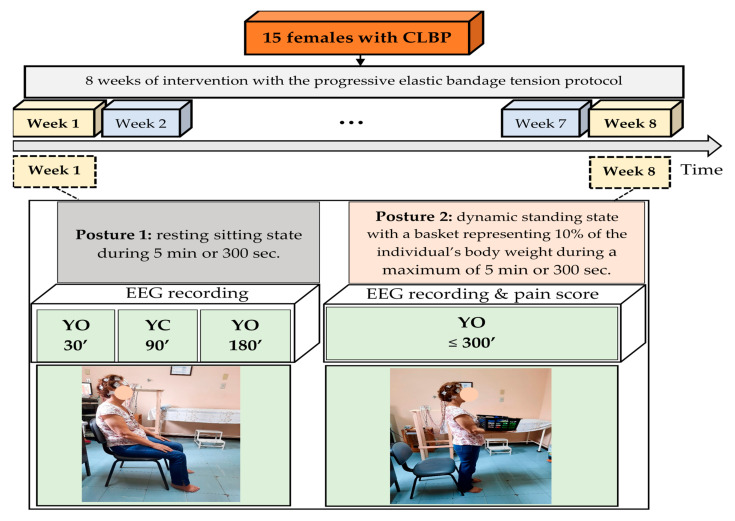
Electroencephalogram (EEG) experimental protocol. CLBP: Chronic low back pain; EEG: electroencephalography; YO: eyes open; YC: eyes closed.

**Figure 3 jfmk-10-00338-f003:**
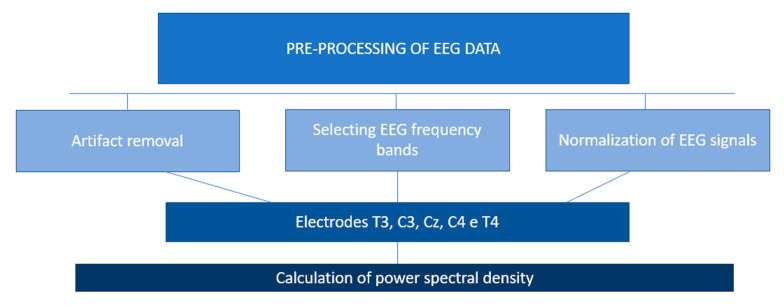
Methodological steps for pre-processing and estimating relative PSD. EEG: Electroencephalography.

**Figure 4 jfmk-10-00338-f004:**
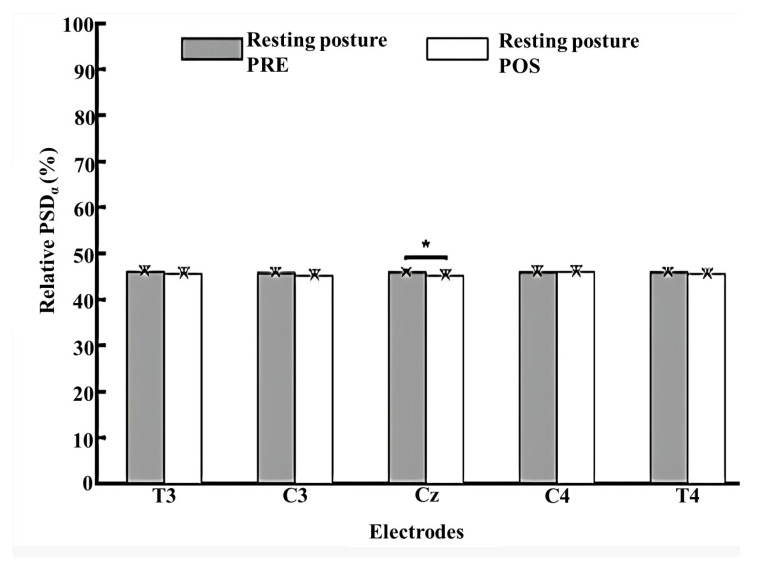
Comparison of relative PSD in the alpha band for five electrodes including T3, C3, Cz, C4, and T4 at pre- and post-intervention (participants at rest). Statistically significant differences (*) were found between data collected from the Cz and T3, C3, C4, and T4 electrodes (*p* < 0.05). PSD: Power spectral density.

**Table 1 jfmk-10-00338-t001:** Characterization of the sample in the first and eighth week of the intervention.

Volunteers	Pre-Intervention		Post-Intervention					
	Week 1		Week 8					
	Resting Posture	Dynamic Standing Posture	Resting Posture	Dynamic Standing Posture	*p*-Value	Duration of Pain (Pre-Intervention)	Age (Years)	BMI (kg/m^2^)
P1	7	7	5	5	0.0001 *	±5 months	20	26.49
P2	7	7	4	4		±3 months	31	23.14
P3	7	7	0	0		±3 months	49	20.9
P4	8	8	0	0		±4 months	40	25.43
P5	7	7	4	4		±4 months	35	20.76
P6	7	7	0	0		±4 months	30	23.05
P7	8	8	0	0		±3 months	31	23.94
P8	8	8	0	0		±6 months	40	28.01
P9	7	7	0	0		±5 months	28	23.39
P10	7	7	0	0		±5 months	41	27.04
P11	8	8	0	0		±4 months	40	20.02
P12	7	7	0	0		±6 months	41	27.61
P13	8	8	5	5		±6 months	42	25.25
P14	8	8	0	0		±4 months	44	26.53
P15	7	7	1	1		±3 months	33	20.9

BMI: Body Mass Index; *: *p*-value < 0.05.

**Table 2 jfmk-10-00338-t002:** Physical disability, beliefs, and fears in relation to movement pre- and post-intervention.

	Pre-Intervention	Pos-Intervention	*p*
*Physical Disability, Beliefs and Fears about Movement*			
ODI (%)	36.80 (7.58)	17.53 (7.64)	0.0001 *
FABQ-Phy (points)	18.50 (6.95)	15.85 (7.81)	0.19
FABQ-Work (points)	32.71 (18.46)	23.71 (18.59)	0.01 *

Represents * *p*-value < 0.05, which is considered statistically significant. Abbreviations: ODI: Oswestry Lumbar Disability Index; FABQ: Fear Avoidance Beliefs Questionnaire; Phy: Physical; Work: Labor.

## Data Availability

The data presented in this study are available on request from the corresponding author due to restrictions (e.g., privacy, legal or ethical reasons).

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
