# Peer review of "Evaluating the Effects of a Progressive Kinesiotaping Treatment Protocol on Chronic Low Back Pain in Women Using Electroencephalography"

_jfmk, 2025, doi:10.3390/jfmk10030338_

Round 1

Reviewer 1 Report

Comments and Suggestions for Authors

General Assessment
This study addresses an important topic — the relationship between Kinesiotaping (KT) and cortical modulation in women with chronic low back pain. The attempt to integrate EEG measures with functional outcomes is interesting and aligns with the growing interest in understanding central mechanisms underlying chronic pain and its modulation through physical interventions. However, there are significant methodological, reporting, and language issues that limit the interpretability and replicability of the findings in its current form.

Strengths

  • The study combines subjective measures (pain, disability, fear-avoidance) with an objective neurophysiological measure (EEG power spectral density).
  • The progressive KT protocol is described in operational detail, including tape placement and tension adjustments.
  • Ethical procedures, inclusion and exclusion criteria are clearly stated.

Major Concerns

Experimental Design and Control

  • Lack of Control Group: The absence of a control or placebo group makes it impossible to determine whether observed effects are due to KT, placebo effects, natural recovery, or the passage of time. A case series design cannot infer causality for neurophysiological outcomes.
  • Confounders: Although within-subject comparisons (pre/post) were done, there is no mention of controlling for potential confounders such as daily activity levels, hormonal fluctuations (especially given the inclusion of only non-menopausal women), or placebo effects.

Recommendation: If a control group is not feasible, the authors must clearly acknowledge this limitation in the Discussion and interpret the findings with appropriate caution.

Language and Style

  • Many sentences are unnecessarily long, with inconsistent punctuation. Ideas are sometimes repeated or phrased redundantly.
  • Some word choices are inaccurate (e.g., injured instead of abnormal — clarify intended meaning).

Clarity and Structure of the Methods Section

The Methods section is overly long yet unclear in key areas:

  • Redundant Sample Information: Eligibility details are repeated in multiple places.
  • Data Analysis Placement: The statistical analysis appears abruptly in the middle of the Methods. It should be moved to the end, after clearly describing how subjective and objective variables were processed.
  • Missing Subsection: Add a clear Data Analysis subsection that describes how all data (subjective questionnaires and EEG) were processed and analyzed.
  • EEG PSD Time Windows: The description of the EEG segments used is confusing. For the resting (seated) task, the baseline and the last three minutes must be consistently described and justified. For the standing task, the Results must clearly report how the segments were selected.
  • Equation Formatting: The Fourier transform equation (Equation 2) is confusing, with corrupted symbols. Rewrite the equations using standard, correctly formatted notation.
  • Normalization: The baseline normalization procedure should be described concisely, with appropriate references to justify this approach.
  • Recording Duration Variation:
    • Table 3 (Results section) shows substantial variation in recorded EEG duration for the dynamic standing task — ranging from just over 1 minute (P14) to over 10 minutes (P1).
    • This major variability affects data comparability. The decision to analyze only the first 60 seconds for all participants must be clearly described in the Methods section, not only in the Results.
    • Clarify whether this truncation applied only to the standing task or also to the resting task. If the full 5 minutes of resting data were used but only 60 seconds of standing, discuss the implications for comparing conditions.
    • Explain the potential impact of short recording windows on PSD estimates — 60 seconds may be insufficient for stable estimates, especially in lower-frequency bands.
    • Describe whether any checks for stationarity or artifact-free data within the selected 60 seconds were performed.
  • Preprocessing Details:
    • Specify whether the IIR Butterworth filter was applied with zero-phase filtering to avoid phase distortion.
    • Clarify the window type (e.g., Hamming, Hann) and the exact overlap percentage.
    • Justify the baseline window (first second or first 30 seconds) — this appears inconsistent and should be explained.
    • Describe how outliers or artifacts were handled — were entire epochs excluded or corrected/interpolated?
    •  
  • Overall Structure:
    The Methods section would benefit greatly from clearer subheadings: Participants, Assessments/Insruments, Procedure, Intervention, Data Acquisition / Recording (EEG Recording), Data Analysis (e.g. EEG Preprocessing, Statistical Analysis.

Results Reporting

  • Standing Task: The Methods state that EEG was recorded during both seated and standing load-bearing tasks, but the abstract and Results do not clearly report findings for the standing task. These results must be included or explained if not analyzed.
  • Table 3: If Table 3 only shows the size/duration of EEG recordings and does not contribute to interpreting results, it should be removed or justified more clearly.

Specific Comments

Introduction

  • Line 42–44: Correct according to the IASP definition: “An unpleasant sensory and emotional experience associated with, or resembling that associated with, actual or potential tissue damage.”
  • Line 65: Define what is meant by pain system. There is no anatomical pain system; consider rephrasing to pain pathways or nociceptive system.
  • Line 69: Clarify alterations in brain function and structure — compared to what? State clearly that this means relative to healthy controls or pain-free individuals.
  • Lines 99–103: Provide an appropriate citation to support this statement.

Methods

  • Line 283: Fix the Fourier transform equation (Equation 2) to ensure standard notation.
  • The normalization procedure should be more concise and supported by references.

Discussion

  • Lines 361–369: Revise statements about cortical regions:
    • Cz primarily reflects activity from sensorimotor regions (S1, M1), which contribute to the sensory-discriminative dimension of pain.
    • The insula and anterior cingulate cortex (ACC) are indeed parts of the pain matrix, but EEG source localization to these regions requires specific source modeling methods (e.g., sLORETA, dipole fitting).
    • A single Cz electrode does not selectively measure insular or ACC activity — revise the interpretation accordingly.

Reviewer 2 Report

Comments and Suggestions for Authors

I greatly appreciate the opportunity to review this important work. It is undoubtedly necessary to identify the functional brain changes generated by physical therapist interventions.

Abstract.
The Oswestry Disability Index is listed in the results section, but it is not listed in the methods section of the abstract. I suggest integrating it as an assessment alongside the Fear Avoidance Beliefs Questionnaire and Numerical Pain Intensity Scale.
In the conclusions, I suggest integrating all variables and their effects. That is, the improvement in pain intensity, disability, and FABQ-work. I also note that there were no significant changes for FABQ-Phy.

Introduction.
In general terms, the information is extremely relevant. However, it should be improved through a more coherent and cohesive writing sequence. For example, they begin by stating the definition of pain (first paragraph), then mention lower back pain (second paragraph), then return to discussing pain with a focus on chronic pain (third paragraph), then provide information on acute pain (fourth paragraph), return to discussing chronic pain (fifth paragraph), then provide information on KT, which is an intervention (sixth and seventh paragraphs), and return to discussing acute and chronic pain (eighth paragraph). It's a back-and-forth of ideas. I suggest trying to limit the number of paragraphs (preferably 6 in total) and including a central idea and several supporting supporting ideas in each paragraph. This would allow for a better connection of the information and would be much clearer for the reader.
Furthermore, there are too many sentences without their citations. I suggest adding the citation after each sentence for greater transparency of the information.

Line 66: CNS not previously described
Line 96: EEG not previously described

Lines 120-122: The objective states that they will analyze the effects of KT on the somatosensory cortex, but it is not explicit whether it is on the structural and functional aspects. Furthermore, the objective does not consider other variables such as pain intensity, disability, and fear-avoidance beliefs. I suggest "To analyze the effects of kinesiotaping on somatosensory cortex function, measured with EEG, and pain intensity, disability, and fear-avoidance beliefs in women with chronic low back pain." If they consider this, they should also change it in the abstract and discussion.

Materials and Methods

I think it's important to separate the evaluations and intervention protocol into two different sections within the methodology. They're mixed together and don't make for easy reading.

Figure A: I believe the image does not fully represent the correct positioning of the KT, as the gluteal folds are likely more than 5 cm from the posterior internal iliac spine.

Figure B: This image does not fully represent what was done, as it does not specify that the pain, fear-avoidance, and disability scales were assessed in week 1. I suggest including an image with everything done and the timing of each step.

The data analysis section should be included at the end of the methodology. Furthermore, in this section, they only indicate that the relative PSD of alpha and beta were integrated with paired t-tests, but the results also indicate that gamma and theta did not show significant changes (in lines 312-313), meaning they were still analyzed. Therefore, I suggest also indicating this in the data analysis section.
Effect sizes were calculated using Cohen's d. However, due to the sample size, this may not be the most appropriate method, and it might be preferable to use hedges' g.

Line 131: There is a 1 after (moderate/severe intensity), which I believe is a reference, but it's not in brackets.

Line 208: NDT is not predefined.

Line 250: There is a 36 after nociception process, which I believe is a quote and is not in brackets.

Results.
In the first paragraph and in Table 1, it would be helpful to provide the average for the 15 participants; this allows for a visualization of the overall change in the sample. This would provide more relevant information to readers.
In Table 1, I suggest indicating: pre-intervention pain intensity week 1 and post-intervention pain intensity week 8.

Figure D. Its resolution should be improved. Furthermore, it appears flattened, and the differences in bar heights are not apparent.

Line 297: Table 2 indicates that it contains the NDT, but it does not appear in the table.

Line 306: EEG is indicated, which I assume is EEG.

Discussion.
The discussion could be much more robust if, for each variable, the results obtained were compared with more studies, ideally prospective studies or systematic reviews with meta-analyses. In this sense, it is not apparent that they compare the effects of KT on pain with previous studies; rather, they discuss prevalence (an aspect that is not the focus of their research) and the possible mechanisms that generate a positive effect.

As in the introduction, I suggest improving the coherence and cohesion. More in-depth research is needed on each variable and the possible mechanisms that underlie the changes due to the use of KT.

Lines 324-326: I suggest discussing effects and improving the wording for clarity.

Lines 332-333: This cannot be stated because they did not include men, so their findings do not confirm what other research has found.

Lines 354-356: I suggest improving the wording of this sentence as it creates confusion.

Lines 361-362: Since the interpretation of pain is not exclusive to one area, I think pointing this out is too risky, especially if it involves a single electrode. To be able to identify specific areas, it is advisable to use at least 64 electrodes and also analyze using sLORETA.

Conclusion.
I suggest also pointing out the findings on fear-avoidance beliefs.
Furthermore, since "reduction in neuronal activity" is mentioned here, I think it is important for the discussion to point out that a reduction in neuronal activity is interpreted as a lower intensity of pain.

Round 2

Reviewer 1 Report

Comments and Suggestions for Authors

The authors have made some improvements in the manuscript, including a more operational description of their EEG analysis and protocol, and acknowledgment of limitations. However, several critical issues from previous reviews remain unresolved. The manuscript continues to suffer from grammatical errors and awkward phrasing, which impede clarity. One example is: Line 123–126: “Thus, this study aimed article was to analyse…” is ungrammatical. Please carefully revise the manuscript for English language and grammar throughout. I recommend professional English editing prior to re-submission.These issues must be thoroughly addressed to ensure scientific clarity, consistency, and methodological soundness.

Major Issues

  1. Interpretation of Findings in Light of Limitations

While the authors now list the study limitations, they do not sufficiently explain how these limitations affect their conclusions. For example:

  • Lack of control group: How do the authors justify interpreting cortical changes (e.g., alpha power decrease at Cz) as an effect of KT, rather than as random variation or placebo effects?
  • Variable EEG recording durations: How does this affect the reliability and comparability of the EEG PSD results? Were any sensitivity analyses performed?

Please explain in the discussion how these limitations may affect the interpretation of your EEG and clinical findings, and justify whether your conclusions remain valid despite them. Be transparent about the exploratory nature of the study.

  1. Incomplete reporting and discussing of dynamic standing task results

In my previous comment, I noted that EEG was recorded during both sitting and standing load-bearing postures. However, only the EEG findings from the sitting condition are described in the Abstract and Discussion. In the revised manuscript, Table 3 presents the duration of the standing task, but its justification remains inadequate.

  • Please clearly report the results from the standing task — for example, include the standing task data in Figure 3 and discuss these findings in the Discussion section.
  • The Abstract currently mentions "sitting and standing" but provides no results for the standing condition. This is misleading and should be corrected or clarified.
  • If Table 3 is retained, its relevance must be explicitly explained in the text. Otherwise, consider removing it if it does not contribute to the interpretation of the findings.

Minor Issues and Technical Corrections

Line 258: Use past tense consistently in Methods. Change “is performed” → “was performed.”

Line 259: Please provide a valid reference to support the assumption that “The first 60 seconds are considered the period of initial activation of the muscle region influenced by the posture during data collection” or rephrase to acknowledge that this is an operational decision rather than a biologically validated standard

Author Response

"Please see the attached file."

Reviewer 2 Report

Comments and Suggestions for Authors

I appreciate the consideration of most of the suggestions provided. However, there are several that were not incorporated, and the reason for not considering them was not justified in the response letter. Despite this, the manuscript has improved, and some suggestions that were not considered could be overlooked, but there are others that could definitely substantially improve the paper.

The abstract seems adequate with the changes made.

Regarding the introduction, I insist that it does not follow a logical sequence. It presented improvements, but it still lacks an adequate order for proper understanding and is too long. Considering this is important so that the reader becomes familiar with the entire structure of the paper.

The objective is still stated from a single variable. If, as a reader, I read the objective, I would expect that the methodology, results, and discussion would only address aspects of the EEG and not other variables such as pain intensity, disability, and fear-avoidance beliefs, which were also considered in the study and therefore should be included in the objective. The objective immediately orients the study's approach and guides its entire work.

In the methodology section, they continue to only mention the alpha and beta frequency bands, even though the results specify alpha, beta, gamma, and theta. For the manuscript to be cohesive, the information must be included in all sections. That is, it should be included in the introduction, methodology, results, and discussion.

Line 317: I believe NPUS is incorrect.

Regarding the discussion, I reiterate the need to compare the results with other studies. Are there studies that indicate that KT produces the same effects you found? Or are there studies that indicate that KT does not reduce pain or improve disability? You should compare these results.

Author Response

"Please see the attached file."
